# Noise-augmented directional clustering of genetic association data identifies distinct mechanisms underlying obesity

Andrew J. Grant[1]*, Dipender Gill[2,3,4,5], Paul D. W. Kirk[1,6], Stephen Burgess[1,7]

**1** MRC Biostatistics Unit, University of Cambridge, Cambridge, United Kingdom, **2** Department of Epidemiology and Biostatistics, School of Public Health, St Mary's Hospital, Imperial College London, London, United Kingdom, **3** Clinical Pharmacology and Therapeutics Section, Institute of Medical and Biomedical Education and Institute for Infection and Immunity, St George's, University of London, London, United Kingdom, **4** Clinical Pharmacology Group, Pharmacy and Medicines Directorate, St George's University Hospitals NHS Foundation Trust, London, United Kingdom, **5** Novo Nordisk Research Centre Oxford, Old Road Campus, Oxford, United Kingdom, **6** Cambridge Institute of Therapeutic Immunology & Infectious Disease (CITIID), University of Cambridge, Cambridge, United Kingdom, **7** Cardiovascular Epidemiology Unit, Department of Public Health and Primary Care, University of Cambridge, Cambridge, United Kingdom

* andrew.grant@mrc-bsu.cam.ac.uk

**Data Availability Statement:** All the data used in this paper are publicly available. Summary statistics for genetic associations with traits were downloaded from:

## Abstract

Clustering genetic variants based on their associations with different traits can provide insight into their underlying biological mechanisms. Existing clustering approaches typically group variants based on the similarity of their association estimates for various traits. We present a new procedure for clustering variants based on their proportional associations with different traits, which is more reflective of the underlying mechanisms to which they relate. The method is based on a mixture model approach for directional clustering and includes a noise cluster that provides robustness to outliers. The procedure performs well across a range of simulation scenarios. In an applied setting, clustering genetic variants associated with body mass index generates groups reflective of distinct biological pathways. Mendelian randomization analyses support that the clusters vary in their effect on coronary heart disease, including one cluster that represents elevated body mass index with a favourable metabolic profile and reduced coronary heart disease risk. Analysis of the biological pathways underlying this cluster identifies inflammation as potentially explaining differences in the effects of increased body mass index on coronary heart disease.

## Author summary

Genome-wide association studies have found many genetic variants that are correlated with traits, particularly complex traits such as body mass index (BMI). However, genetic association data cannot tell us how these variants influence the trait, or whether they influence the trait in the same way. Insight into these questions may be gained by analysing the associations between the variants and other related traits. Variants with similar patterns of associations across a set of traits may be thought to act via similar biological mechanisms.

https://zenodo.org/record/1251813#.X8drUF7gquP (BMI and WHR); http://www.nealelab.is/uk-biobank/ (body fat percentage, SBP, triglycerides, HDL and CRP); https://www.thessgac.org/data (educational attainment); https://ora.ox.ac.uk/objects/uuid:ff479f44-bf35-48b9-9e67-e690a2937b22 (physical activity); https://data.bris.ac.uk/data/dataset/10i96zb8gm0j81yz0q6ztei23d (lifetime smoking score); http://diagram-consortium.org/downloads.html (T2D); http://www.phenoscanner.medschl.cam.ac.uk/ (CHD); https://data.bris.ac.uk/data/dataset/3g3i5smgghp0s2uvm1doflkx9x (cytokines and growth factors). R code for performing the NAvMix clustering algorithm, and for reproducing the simulation results and applied analysis, can be found at https://github.com/aj-grant/navmix.

**Funding:** AJG and SB are supported by a Sir Henry Dale Fellowship jointly funded by the Wellcome Trust and the Royal Society (grant number 204623/Z/16/Z). DG is supported by the British Heart Foundation Research Centre of Excellence (RE/18/4/34215) at Imperial College London and a National Institute for Health Research Clinical Lectureship (CL-2020-16-001) at St. George's, University of London. PDWK is supported by the UK Medical Research Council (MC_UU_00002/13). This research was funded by the NIHR Cambridge Biomedical Research Centre (BRC-1215-20014). The views expressed are those of the authors and not necessarily those of the NHS, the NIHR or the Department of Health and Social Care. For the purpose of open access, the author has applied a CC-BY public copyright licence to any Author Accepted Manuscript version arising from this submission. The funders had no role in study design, data collection and analysis, decision to publish, or preparation of the manuscript.

**Competing interests:** I have read the journal's policy and the authors of this manuscript have the following competing interests. DG is employed part-time by Novo Nordisk. The other authors declare no competing interests.

Here we present a new statistical method for grouping genetic variants according to their associations with chosen traits, so that each group represents variants acting on these traits in a distinct way. We apply the method to genetic variants associated with BMI and then study the effects of each of the identified groups of variants on coronary heart disease. We find a group of genetic variants associated with higher BMI and decreased risk of heart disease, which is in contrast to the established overall harmful effect of BMI on heart disease.

## Introduction

In recent years, the number of genome-wide association studies (GWAS) has grown enormously [1]. Such studies provide valuable information linking genetic variants across the human genome to a wide range of traits. What often remain less understood are the underlying mechanisms by which the associated genetic variants affect the traits. Insight into these mechanisms may be gained by investigating the pattern of associations with other related traits: genetic variants that share similar association patterns may be thought to act via similar mechanisms [2]. For example, some genetic variants associated with type 2 diabetes are also associated with obesity related traits such as body mass index (BMI), whereas others are instead associated with traits such as triglycerides, suggesting that the variants influence type 2 diabetes risk via different biological mechanisms [3].

A number of techniques have been implemented to cluster genetic variants based on their associations with traits that are believed to be relevant in informing biological pathways. The traits often include separate risk factors or potential mediators of some disease outcome(s) of interest. A common approach is to use hierarchical clustering, which groups observations based on their distance from each other [4–7]. The number of clusters is then chosen heuristically. Other clustering approaches which have been applied to genetic variant-trait association estimates include fuzzy c-means [6] and Bayesian nonnegative matrix factorization [3]. A related approach which aims to determine distinct components of genetic variant-trait associations uses truncated singular value decomposition [8].

A key characteristic of previously implemented approaches is that they cluster based on the Euclidean distance between vectors of the genetic variant-trait association estimates, defined as the length of the line between the association estimates plotted as points on a graph. However, when trying to determine shared biological mechanisms, a more relevant clustering target is the proportional associations of each genetic variant with the set of traits. If two variants influence a set of related traits via a common mechanism, the genetic associations may differ considerably in magnitude due to one variant having a stronger effect than the other. However, their proportional associations across the traits will be similar for both variants. Equivalent to looking at proportional associations is to consider the direction of the association vector from the origin. That is, in order to distinguish between variants which act via different mechanisms, it is the direction of the association vector rather than its location in space which is of most importance. This is illustrated graphically in Fig 1. Relating similar directions of genetic associations to shared biological mechanisms has been discussed by, for example, Yaghootkar et al. [9], Winkler et al. [2] and Udler et al. [3]. We note that implicit in this definition of mechanism is the assumption that the relationships between the genetic associations with one trait and the genetic associations with each of the other traits are linear.

In this paper we introduce a novel procedure for clustering genetic variants based on their associations with a given set of traits to identify groups with common biological mechanisms.

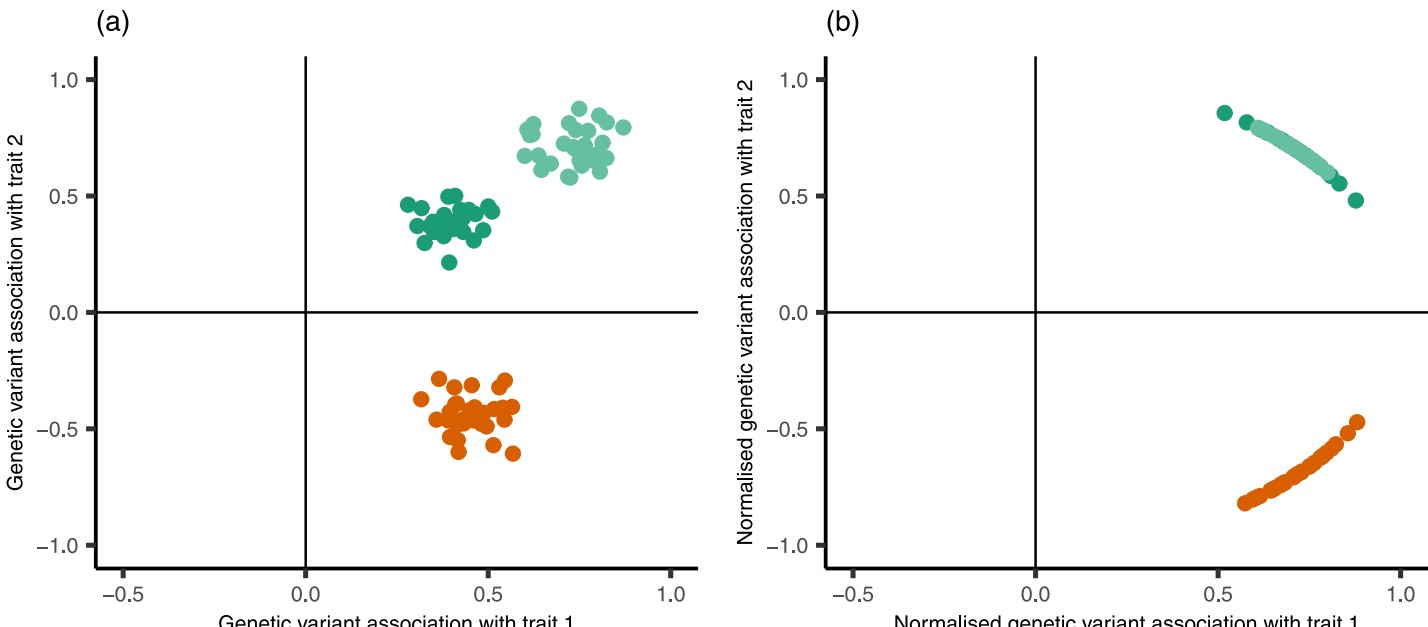

**Fig 1. Illustrative figure showing the difference between clustering based on Euclidean distance compared with direction.** Panel (a) plots 90 simulated points representing genetic associations with two traits. Each point was generated from one of three bivariate normal distributions. Panel (b) plots the normalised genetic associations, representing the proportional association of each genetic variant with respect to the two traits. All points sit on the unit circle. The green points represent genetic variants which are positively associated with each trait by similar magnitudes. The orange points represent genetic variants which are positively associated with trait 1 and negatively associated with trait 2, again by similar magnitudes. Methods based on Euclidean distance such as Gaussian mixture models and hierarchical clustering would consider there to be three clusters, distinguishing between the light and dark green points, as shown in Panel (a). Directional clustering approaches would consider there to be two clusters, grouping the green points in the same cluster. This is shown in Panel (b), where the points are clearly grouped in two separate clusters.

We develop the NAvMix (Noise-Augmented von Mises–Fisher Mixture model) clustering method, which extends a directional clustering approach to include a noise cluster as well as a data-driven method for choosing the number of clusters. The method is shown in a simulation study to perform well in identifying true clusters and to outperform alternative approaches across a range of scenarios. We further apply the procedure to cluster genetic variants associated with body mass index (BMI). We study the downstream effects of the different components of BMI on coronary heart disease (CHD) using Mendelian randomization, which uses genetic variants as instrumental variables to study potential causal effects of a risk factor on an outcome [10, 11]. We identify a BMI increasing cluster of variants associated with a favourable cardiometabolic profile and lower CHD risk. Analysis of the biological pathways which underlie each group of variants suggests that a key difference of this cluster compared with the others is its distinct effect on systemic inflammation. The clustering method demonstrated in this work is thus able to identify distinct pathways underlying complex traits, in turn highlighting specific mechanisms for therapeutic intervention.

## Results

### Overview of the proposed clustering approach

We use a mixture model approach to clustering, which supposes that each observation is a realisation from one of a fixed number of probability distributions. Since we are interested in clustering based on direction of association, we fit a mixture of von Mises–Fisher (vMF)

distributions, which is a distribution characterised by the mean direction of the observations from the origin and a dispersion parameter. A mixture model of vMF distributions has previously been described by Banerjee et al. [12]. We augment this approach by including a noise cluster, in recognition of the fact that not all observed vectors of genetic variant-trait association estimates are expected to fit well within the set of specified distributions. The noise cluster will contain outliers to the specified model, providing robustness to the identification of clusters. Our method of clustering is thus to fit a Noise-Augmented von Mises–Fisher Mixture model (NAvMix).

The NAvMix algorithm outputs a probability for each observation belonging to each cluster based on the given data. Each observation can then be assigned according to which cluster it has the highest probability of membership (referred to as hard clustering). The approach also provides the ability for soft clustering, which is where an observation is assigned to any cluster for which it has a probability of membership over a certain level, so that observations may belong to more than one cluster. Although the algorithm requires a fixed number of clusters to be specified, we repeat the procedure for varying numbers of clusters then chose the final number using the Bayesian Information Criterion (BIC). Full details of the procedure are given in the Methods section.

Let $\hat{\boldsymbol{\beta}}_{j.}$ be the vector of association estimates of genetic variant $j$ with the set of traits under consideration, and let $\hat{\Sigma}_{j}$ be the covariance matrix of this vector. We assume that the genetic variants are independent of each other (that is, no linkage disequilibrium). We also note that the association estimates do not need to have been taken in the same sample, so we can consider sets of associations between genetic variants and any trait for which corresponding GWAS summary statistics are available. Although it is possible to input the raw association estimates into the algorithm, we propose inputting the standardised association estimates, given by $\hat{\Sigma}_{j}^{-1/2}\hat{\boldsymbol{\beta}}_{j.}$ for the $j$th variant. The standardisation means that each element of the input vector is independent and has the same standard error. It thus is able to account for correlation between association estimates. Assuming all genetic associations are estimated with the same sample size for a given trait, this will not distort the direction vector. If there are significant differences between sample sizes used to estimate genetic associations for the same trait, and associations with different traits are on similar scales, the unstandardised association estimates may also be used, possibly as a sensitivity analysis. The first step in the algorithm is to transform each input vector to have a magnitude of one. This is done by dividing each vector by its Euclidean distance from the origin. We shall refer to this as normalisation. The normalised vectors represent the proportional association estimates.

The diagonal elements of the covariance matrices represent the variances of the genetic variant-trait association estimates. The off-diagonal elements represent the covariances between these estimates. If the genetic associations are estimated in separate samples for each trait, these covariances will be theoretically equal to zero. If the association estimates are taken from the same sample, the covariances will still be approximately zero if the traits are independent. If the traits are correlated, an estimate of this correlation is required to estimate the full covariance matrix in the one sample setting. This is easily computed using individual level data (Methods). If published GWAS summary statistics are being used, this information will not always be available. Nonetheless, the simulation study presented in the following section shows the clustering approach still performs well in the case where traits are truly correlated but the correlation estimates are set to zero.

## Simulation results

We performed a simulation study in order to evaluate the performance of the proposed method and to compare it with alternative clustering approaches. We chose two methods for comparison. The first was to fit Gaussian mixture models to the standardised association estimates using the mclust algorithm in R [13]. The method was chosen for comparison because it is a model-based approach that is able to estimate the number of clusters by fitting multiple models and choosing between them using a principled model selection criterion. The second approach used for comparison was to fit Gaussian mixture models using the proportional association estimates. This is a case of model misspecification, since the association estimates after normalisation will not follow Gaussian distributions, even if the association estimates themselves do (see, for example, Fig 1). It thus demonstrates the result of applying a method for clustering based on Euclidean distance to proportional associations. Note that other R packages which implement a form of directional clustering were not used for comparison because they either do not allow for estimation of the number of clusters (for example, skmeans [14], which uses the spherical k-means algorithm) or do not incorporate a noise cluster (for example, movMF [15]), and so performance cannot easily be compared.

We simulated data for genetic variants across six scenarios, where the number of traits (denoted by $m$) was either 2 or 9 and the number of clusters ($K$) was either 1, 2 or 4. In each scenario, each of 80 genetic variants were associated with one of $K$ latent factors, representing the different clusters. Each trait was a function of these latent factors, 20 additional noise genetic variants, and random variation of which a proportion, determined by the parameter $\gamma$, came from a shared unmeasured confounding variable. The $\gamma = 0$ case represents uncorrelated traits, however it also proxies the scenario where the traits may be correlated but measured in separate, non-overlapping samples. Increasing values of $\gamma$ therefore demonstrate the effect of increased trait correlation and/or sample overlap. We applied NAvMix in two ways. In the first, the off-diagonal entries of the covariance matrices were set to zero. In the second, the estimated trait correlation from individual level data was incorporated into the procedure, so the full estimated covariance matrices were used. In the primary simulation study presented here, the genetic variant-trait associations were estimated in a single sample of 20 000 individuals. S1 Text also presents the results of a simulation study where the sample sizes for each trait differed. Full details of the simulation parameters are given in the Methods section.

We evaluated the performance of each method using four measures: the adjusted Rand index; the silhouette coefficient; the mean number of clusters estimated; and the mean number of observations assigned to the noise cluster. The adjusted Rand index is a similarity measure between the true and estimated cluster memberships, and shows how well each method allocated the observations [16, 17]. The closer to 1, the closer the estimated cluster membership is to the truth. The silhouette for an observation is based on its closeness to other observations within its cluster and its separation from observations outside its cluster [18]. A higher value indicates that the observation fits well within its allocated cluster. We define the distance between two observations as the distance along the surface of the unit sphere after normalising, and we define the silhouette coefficient as the mean silhouette of all observations, with a higher silhouette coefficient indicating better formed clusters. Fig 2 shows boxplots of the adjusted Rand index for each method and scenario. Boxplots of the silhouette coefficients are shown in Fig A in S1 Text. Table 1 shows the mean number of clusters estimated and the mean size of the noise cluster for each method and scenario.

NAvMix performed very well in terms of allocating the observations to the correct clusters, with a median adjusted Rand index above the mclust approaches in nearly all scenarios. It similarly outperformed with respect to the silhouette coefficient, and selected, on average,

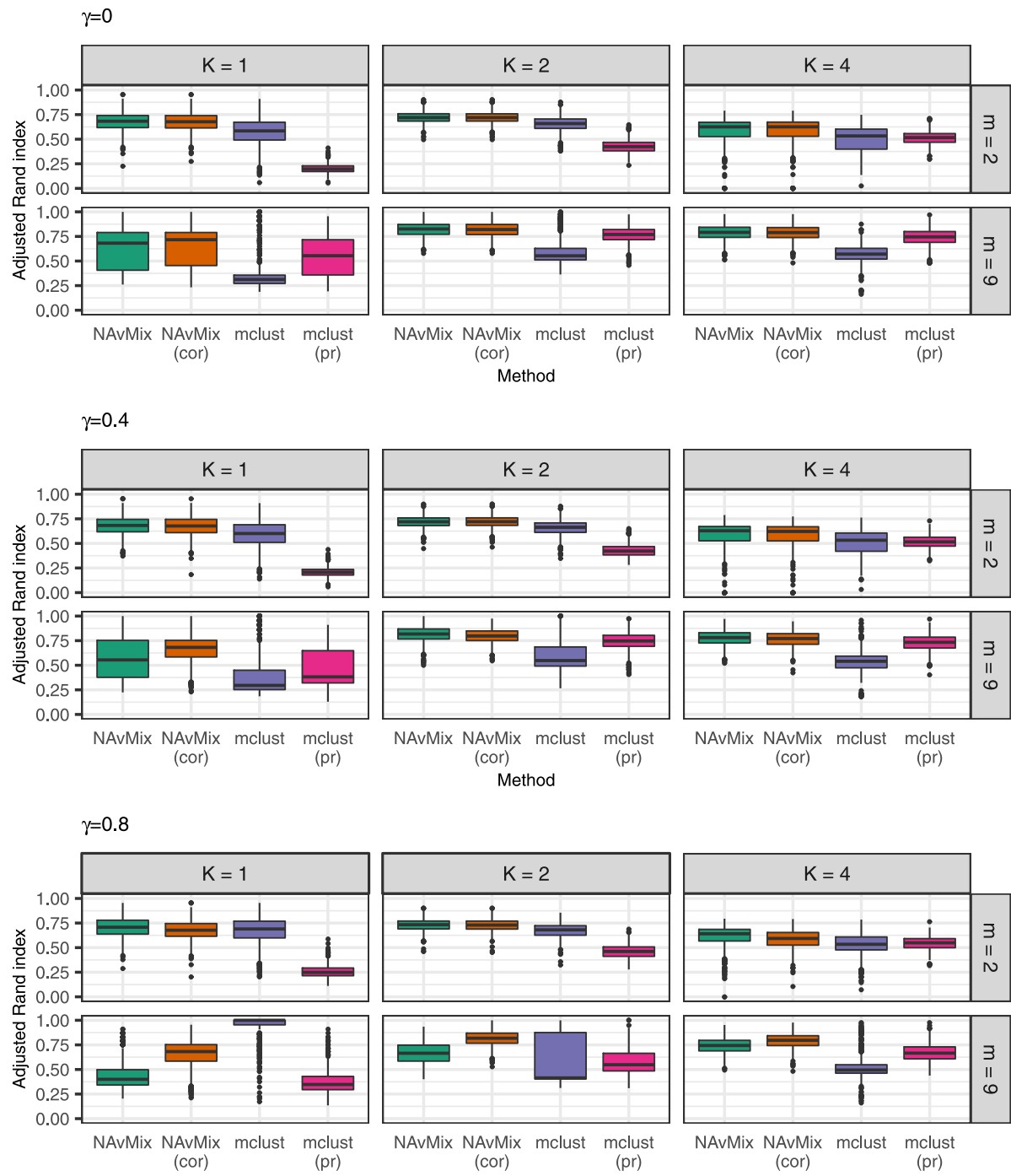

**Fig 2. Comparison of methods in the simulation study.** Boxplots of the adjusted Rand index for each scenario using NAvMix, NAvMix incorporating trait correlation estimates (cor), mclust, and mclust using proportional associations (pr).

a number of clusters closer to the true number. The mclust algorithm tended to overestimate the number of clusters, particularly when there were no truly distinct clusters (that is, in the $K = 1$ scenarios). The exception was when the traits were highly correlated (with $\gamma = 0.8$), where NAvMix tended to select too many clusters. However, incorporating the trait

**Table 1. Mean number of clusters estimated and mean number of observations allocated to the noise cluster for each simulated scenario using NAvMix, NAvMix incorporating trait correlation estimates (cor), mclust, and mclust using proportional associations (pr).** The true number of variants in the noise cluster is 20.

| $\gamma$ | Number of traits ($m$) | Number of clusters ($K$) | Number of clusters | | | | Number of noise variants | | | |
|---|---|---|---|---|---|---|---|---|---|---|
| | | | NAvMix | NAvMix (cor) | mclust | mclust (pr) | NAvMix | NAvMix (cor) | mclust | mclust (pr) |
| 0 | 2 | 1 | 1.00 | 1.01 | 1.19 | 7.09 | 19.88 | 19.93 | 18.17 | 8.93 |
| | | 2 | 2.00 | 2.00 | 2.07 | 8.08 | 17.63 | 17.66 | 16.79 | 6.61 |
| | | 4 | 3.66 | 3.67 | 3.45 | 8.35 | 14.09 | 13.95 | 13.53 | 6.55 |
| | 9 | 1 | 1.42 | 1.29 | 3.41 | 1.52 | 23.83 | 24.77 | 19.69 | 25.98 |
| | | 2 | 2.04 | 2.03 | 4.99 | 2.09 | 26.17 | 26.46 | 19.88 | 28.34 |
| | | 4 | 4.17 | 4.11 | 4.19 | 4.09 | 24.93 | 25.68 | 19.34 | 28.39 |
| 0.4 | 2 | 1 | 1.00 | 1.00 | 1.20 | 6.93 | 20.18 | 20.41 | 18.11 | 9.55 |
| | | 2 | 2.00 | 2.00 | 2.06 | 8.07 | 17.63 | 17.62 | 16.75 | 6.75 |
| | | 4 | 3.66 | 3.61 | 3.47 | 8.32 | 13.10 | 15.71 | 13.81 | 6.54 |
| | 9 | 1 | 1.56 | 1.14 | 3.30 | 1.73 | 24.00 | 26.86 | 19.41 | 26.03 |
| | | 2 | 2.08 | 2.03 | 4.33 | 2.21 | 26.59 | 27.40 | 19.15 | 28.71 |
| | | 4 | 4.18 | 4.02 | 2.88 | 4.09 | 25.65 | 27.39 | 18.37 | 28.93 |
| 0.8 | 2 | 1 | 1.01 | 1.01 | 1.22 | 6.52 | 21.20 | 22.27 | 18.18 | 10.95 |
| | | 2 | 2.00 | 2.00 | 2.04 | 8.01 | 17.91 | 17.86 | 16.50 | 6.68 |
| | | 4 | 3.79 | 3.33 | 3.38 | 8.13 | 12.12 | 22.60 | 12.70 | 7.80 |
| | 9 | 1 | 1.97 | 1.13 | 1.11 | 2.17 | 23.85 | 27.04 | 19.28 | 25.40 |
| | | 2 | 3.49 | 2.04 | 1.98 | 4.22 | 24.52 | 27.01 | 18.42 | 25.67 |
| | | 4 | 4.44 | 4.00 | 2.34 | 5.60 | 26.90 | 27.12 | 18.68 | 28.07 |

correlation estimates in NAvMix improved the performance in these cases. Note that when $K = 4$, one of the clusters had only 10 genetic variants. Nonetheless, NAvMix still selected close to 4 clusters, on average, and had higher median adjusted Rand indices and silhouette coefficients than the mclust approaches. Other than in the scenarios with both a higher number of traits ($m = 9$) and high trait correlation ($\gamma = 0.8$), there was not a big difference in the results between using NAvMix with and without trait correlation estimates. This suggests that, unless there is substantial trait correlation or sample overlap, the procedure is robust to missing these estimates. Incorporating trait correlation becomes more important as the number of traits increases and the number of true clusters decreases. Finally, mclust tended to allocate fewer observations to the noise cluster than NAvMix, particularly in the lower dimensional ($m = 2$) settings.

We repeated the analysis on the same simulated datasets but where the genetic variants were filtered such that only those which associated with at least one trait at genome-wide significance were included. This greatly improved the performance of NAvMix in the highly correlated trait scenarios (see Figs B and C and Table A in S1 Text). In the simulation scenarios where the sample sizes differed, the results were similar to those of the primary simulation study (see Figs D and E and Table B in S1 Text). In these scenarios, the various sample sizes were up to five times different, suggesting that the procedure is robust to reasonably large differences in sample sizes used for each trait.

## Clustering BMI associated genetic variants

We applied our procedure to cluster BMI associated genetic variants identified by the GWAS of Pulit et al. [19]. We considered genetic variants associated with BMI at a p-value $< 5 \times 10^{-8}$ and pruned at $r^2 < 0.001$. The clustering was performed in relation to the genetic associations

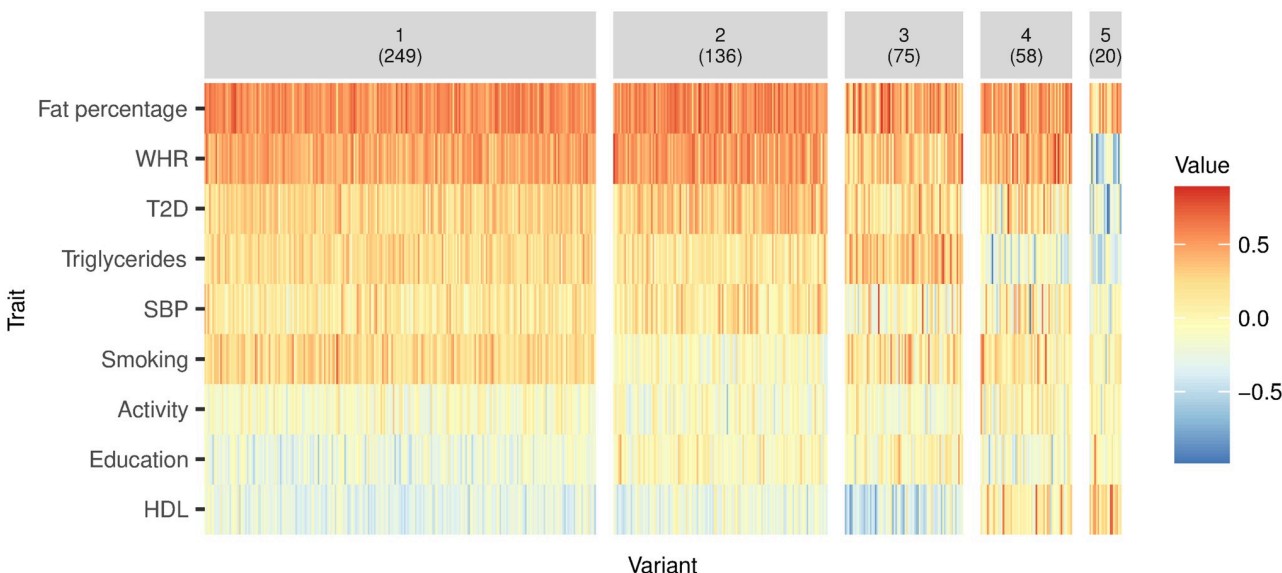

**Fig 3. Heat map showing the association estimates of the BMI associated genetic variants with each trait by cluster.** The association estimates were first standardised by dividing by their standard errors, then normalised so that the vectors of association estimates for each variant have magnitude one. Thus, the values shown represent the proportional association estimates for each genetic variant on the set of traits. The value in parentheses underneath each cluster label is the number of variants in the respective cluster.

with nine traits: body fat percentage; systolic blood pressure (SBP); triglycerides; high-density lipoprotein cholesterol (HDL); educational attainment; physical activity; lifetime smoking score; waist-to-hip ratio (WHR); and type 2 diabetes. These are lifestyle or cardiometabolic traits which have previously been shown to be related to BMI and which may offer insight into the pathways to downstream effects of BMI such as CHD [20, 21]. The genetic association estimates with these traits were all obtained from publicly available GWAS summary statistics (Methods). We clustered the 539 genetic variants that were available across all datasets. The full list of genetic variants and their allocated cluster, along with their probabilities of membership for each cluster, is given in S1 Table.

Five clusters were identified, with 1 genetic variant allocated to the noise cluster. Fig 3 shows a heat map of the proportional genetic association estimates with each trait by cluster and Fig 4 plots the means of each fitted vMF distribution, representing the proportional associations for an observation at the centre of each cluster. The largest four clusters, labelled Clusters 1–4, contain genetic variants with very similar positive average proportional associations with fat percentage, WHR and type 2 diabetes. Variants in Cluster 3 have close to zero average association with SBP, whereas those in Clusters 1, 2, and 4 have positive average association with SBP. Variants in Cluster 2 have close to zero average association with smoking, whereas those in Clusters 1, 3 and 4 have positive average association with smoking. Variants in Cluster 4 have positive average association with HDL and negative average association with triglycerides, in contrast with those in Clusters 1–3.

Cluster 5 contains 20 genetic variants. These variants, on average, are positively associated with HDL and negatively associated with SBP, triglycerides, WHR and type 2 diabetes. These variants also have close to zero average association with smoking, physical activity and education, as well as weaker positive association with fat percentage compared with the other four clusters.

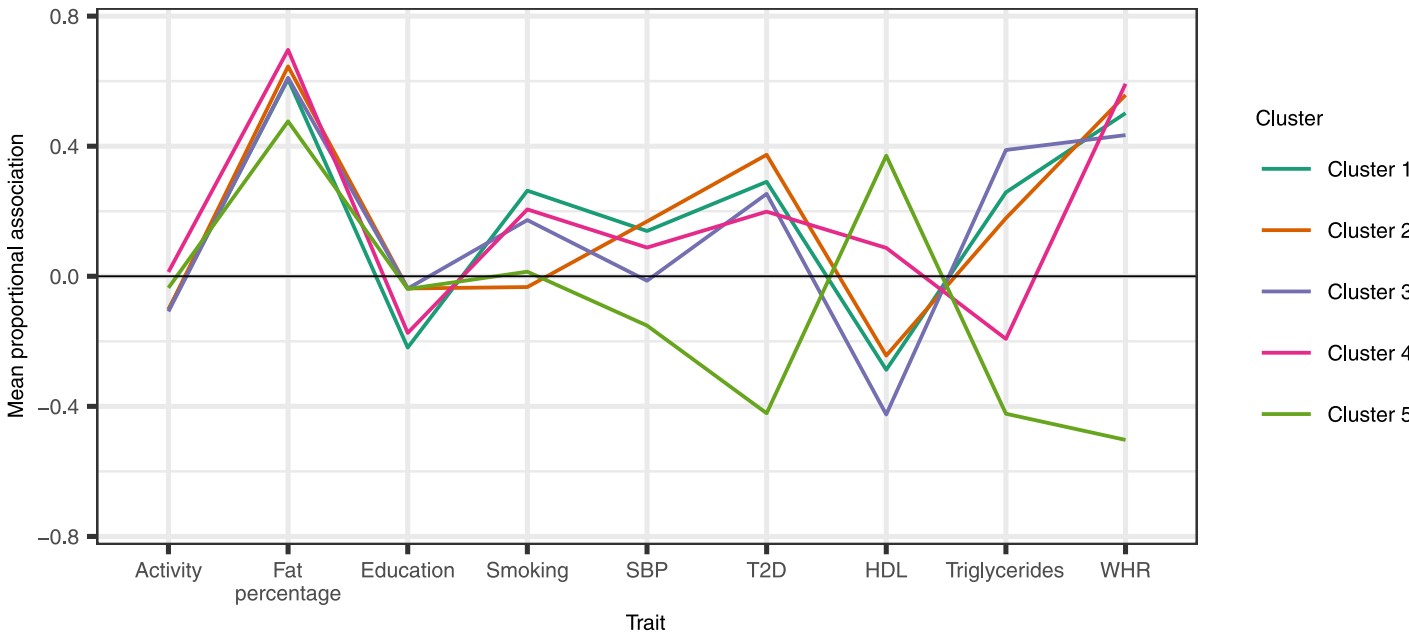

**Fig 4. Parallel plot of the mean vector of the fitted von Mises–Fisher distribution for each cluster.** The plotted points represent the standardised proportional association with each trait for an observation at the centre of each cluster.

## Mendelian randomization estimates of the effect of BMI on CHD

Mendelian randomization has previously suggested that BMI has a positive causal effect on CHD risk using as instruments 94 genetic variants identified by Locket et al. [22][23]. We applied two-sample Mendelian randomization [24] using as instruments the set of BMI associated genetic variants which were used for clustering, as well as separately using the sets of variants for each cluster in turn (Methods). As well as applying the inverse-variance weighted (MR-IVW) method [25], we also performed as sensitivity analyses the MR-Median method [26], the Contamination Mixture (MR-ConMix) method [27] and the MR-PRESSO method [28]. Each of these methods provides a valid test for the causal null hypothesis under different sets of assumptions (Methods).

Fig 5 shows scatterplots of the genetic association estimates with BMI against their association estimates with CHD risk for each set of instruments considered, as well the results of the Mendelian randomization analyses. When using the full set of genetic variants as instruments, the results suggest a positive effect of increased BMI on CHD risk, with an estimated odds ratio (OR) from MR-IVW of 1.50 (95% confidence interval of 1.40–1.62) per 1 standard deviation increase in genetically predicted BMI. All sensitivity analyses gave similar estimates. This is in line with the results of Larsson et al. [23]. A similar result was obtained using the largest two clusters, with an estimated OR of 1.83 (1.68–2.00) using Cluster 1 and of 1.54 (1.38–1.72) using Cluster 2. When using the Cluster 3 genetic variants as instruments, the estimate attenuated toward the null, with an estimated OR of 1.22 (0.99–1.50). When using Cluster 4 genetic variants as instruments, there was no evidence that increased BMI is associated with CHD risk, with an estimated OR of 0.94 (0.69–1.29). When using Cluster 5 genetic variants as instruments, the results suggest a decrease in CHD risk from increased BMI, with an estimated OR of 0.34 (0.19–0.64). Note that the MR-Egger intercept test [29] did not show evidence of directional pleiotropy in any of these analyses (see Table C in S1 Text).

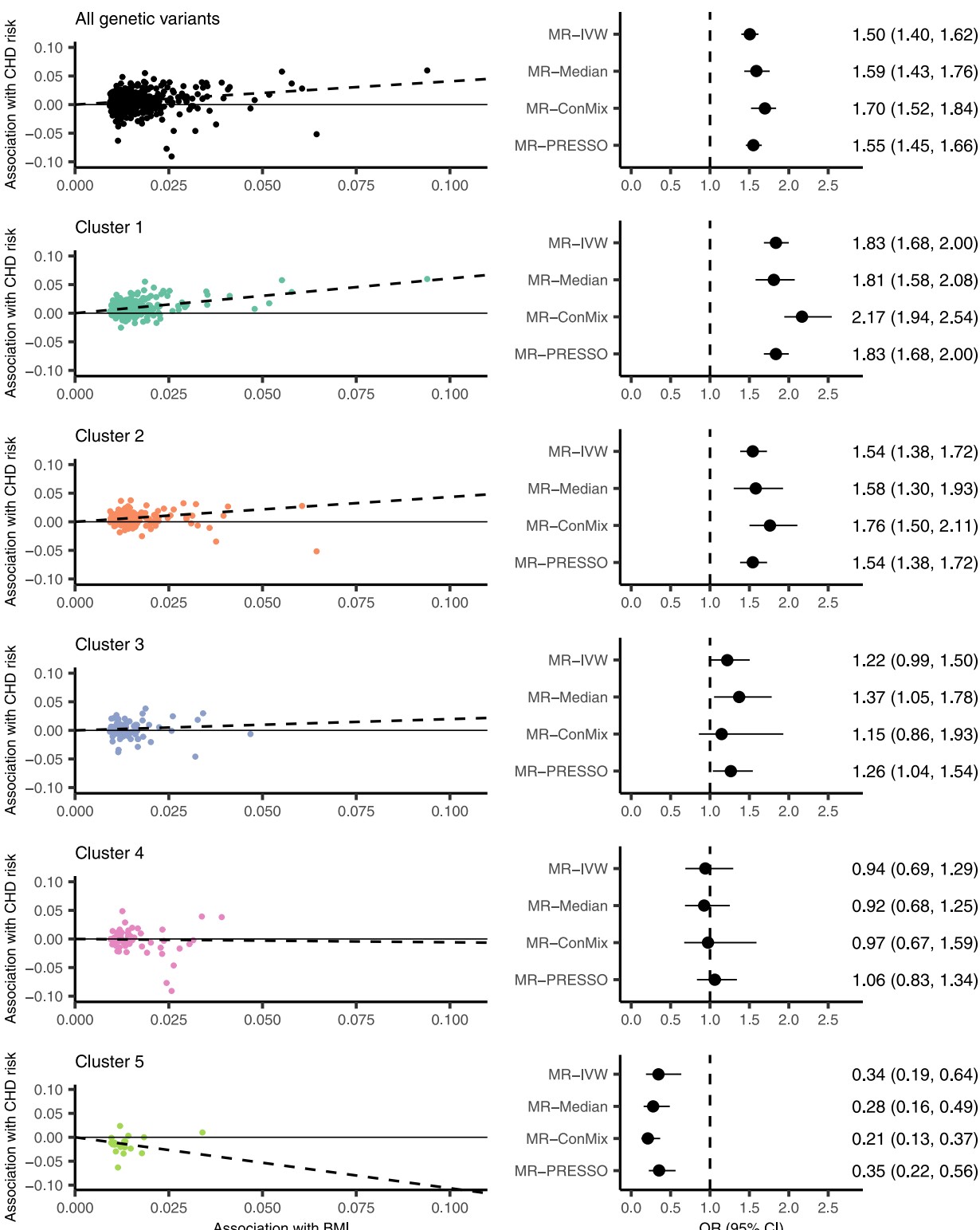

**Fig 5. Results from the Mendelian randomization analyses of the effect of BMI on CHD.** Scatterplots are of the associations of each genetic variant with BMI (standard deviation units) and the log odds ratio of CHD risk. The slopes of the dotted lines are the MR-IVW estimates for the respective cluster. Forest plots show the estimates and 95% confidence intervals from Mendelian randomization, for all genetic variants and for each cluster. Mendelian randomization estimates represent the change in odds ratio of CHD risk per 1 standard deviation increase in genetically predicted BMI. The dotted lines indicate an odds ratio of 1.

## Exploring the biological pathways of clusters of BMI associated variants

We conducted gene set analysis on the BMI associated variants using the Functional Mapping and Annotation Platform [30] in order to examine the biological pathways relating to each cluster. The variants were mapped to genes based on positional and eQTL mappings, which were in turn tested for enrichment in gene sets from various pathway databases (Methods). A number of distinct patterns emerge: Cluster 1 variants are associated with pathways related to cell division and differentiation; Cluster 3 variants with pathways related to cellular signalling; Cluster 4 variants with pathways related to lipid metabolism; and Cluster 5 variants with pathways related to inflammation. Cluster 2 variants were not found to be significantly enriched with any of the tested pathways. The full set of pathways associated with the mapped genes is given in S2 Table.

The role of Cluster 5 variants in inflammation is of particular interest given its proposed relation to favourable adiposity. In order to confirm the role of these variants in inflammation, we conducted a Mendelian randomization analysis to examine the association of genetically predicted BMI, using all variants and each cluster separately, with C-reactive protein (CRP), a measure of systemic inflammation (Methods). The results from the MR-IVW method are shown in Fig 6. When using all variants as instruments, MR-IVW estimated an increase in CRP of 0.44 standard deviations (95% confidence interval of 0.38–0.50) per standard deviation increase in genetically predicted BMI. The results when using Clusters 1–4 as instruments were in line with this. However, there was no evidence that the component of BMI predicted by Cluster 5 variants is associated with CRP (MR-IVW estimate of 0.01, 95% confidence interval of -0.24–0.27). These findings were supported in sensitivity analyses (see Fig F in S1 Text).

To further explore the pathways by which the various clusters affect inflammation, we performed separate Mendelian randomization analyses with the 41 cytokines and growth factors studied by Ahola-Olli et al. [31] and Kalaoja et al. [32] as outcomes (see Table D in S1 Text for the full list of cytokines and growth factors considered). Fig 7 shows the MR-IVW estimates for each cluster and outcome. There was evidence of variation in the effects of BMI predicted

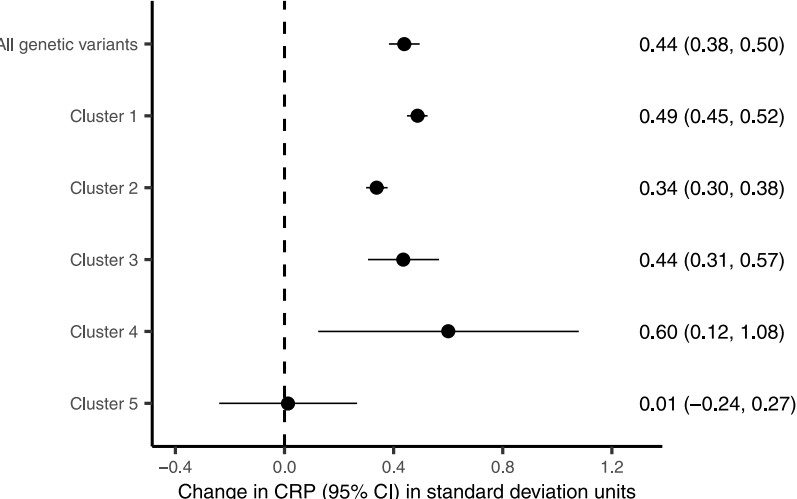

**Fig 6. Results from the Mendelian randomization analyses of the effect of BMI on CRP.** MR-IVW estimates and 95% confidence intervals of the association of genetically predicted BMI with CRP, for all genetic variants and for each cluster. The estimates represent the change in CRP in standard deviation units per 1 standard deviation increase in genetically predicted BMI. The dotted line indicates no association between genetically predicted levels of CRP and BMI.

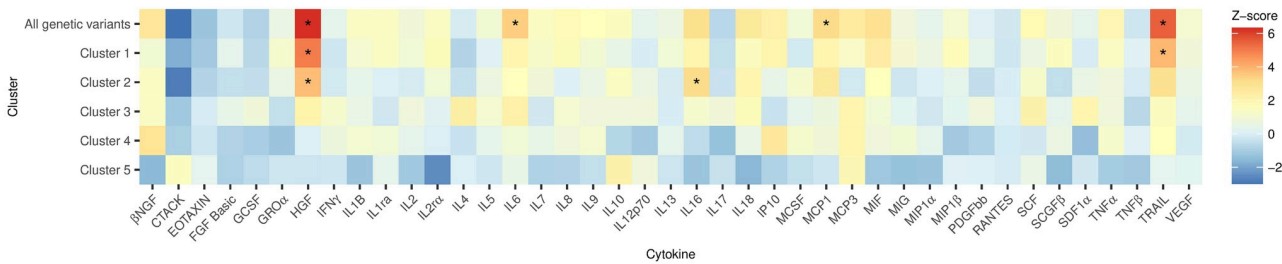

**Fig 7. Results from the Mendelian randomization analyses of the effect of BMI on cytokines and growth factors.** MR-IVW estimates (expressed as Z-scores, i.e. estimate divided by its standard error) for the association of genetically predicted BMI with 41 cytokines and growth factors. Values denoted with * have a p-value less than 0.05/41.

by Cluster 5 variants on the cytokines compared with the effects of BMI predicted by the other clusters. For a number of inflammatory traits, such as hepatocyte growth factor (HGF) and TNF-related apoptosis inducing ligand (TRAIL), BMI predicted by Cluster 5 variants showed a weaker association than the other clusters. In some cases, such as for monocyte chemotactic protein-1 (MCP1), the MR-IVW estimates using Cluster 5 variants were in the opposite direction to the other clusters. These results were supported in sensitivity analyses (see S3 Table).

## Discussion

In this paper we have presented a procedure for clustering genetic variants based on their associations with a given set of traits using the NAvMix method. The method uses a directional clustering algorithm to distinguish between genetic variants based on their proportional associations with the traits. Since it is a model-based clustering approach, it has many advantages over current methods that are employed for clustering genetic variants based on trait associations, such as a data-driven method for choosing the number of clusters and the ability to use soft clustering. The inclusion of a noise cluster provides robustness to outliers, offering greater confidence in the identified clusters. A simulation study showed the method performs well in a range of settings, and that it outperformed alternative clustering approaches in assigning observations based on proportional associations. Importantly, the method did not identify false positive clusters in the simulation setting when no true clusters existed in the data, in contrast to the other methods considered.

The application to clustering BMI associated genetic variants identified five clusters, suggesting that genetic predictors of BMI can be broken down into five separate mechanisms based on their associations with the traits considered. Interestingly, variants in Clusters 1 and 2 were similar in their average associations across each of the traits considered with the exception of smoking, where Cluster 2 had close to zero association. One possible explanation for this is that these variants differ according to some addictive behaviour related mechanism. However, no such pathways were identified in the gene set analysis for Cluster 1. This suggests that some other mechanism may be driving this change, although further analysis is required to identify what this may be.

Mendelian randomization analyses provided evidence that the different pathways affecting BMI have different downstream effects on CHD risk. When using as instruments the set of genetic variants in Clusters 1 and 2, the Mendelian randomization estimate of BMI on CHD risk was positive, in line with the established overall effect of increased BMI. When using as instruments the set of variants in Cluster 3, the estimate was still positive but attenuated to the null. The main difference between this cluster and Clusters 1 and 2 is that the variants do not, on average, associate with increased SBP. Previous evidence suggests that increased SBP is a

downstream consequence of increased BMI [33], and has also been shown to have a causal effect on CHD [27]. Our results therefore support that the genetically predicted component of BMI that does not associate with increased SBP has a lower positive effect on CHD risk. However, there is still evidence of a positive causal effect, suggesting there are other mechanisms by which increased BMI may increase CHD risk [34].

When using as instruments the set of genetic variants in Cluster 4, which have average associations with increased HDL and decreased triglycerides, Mendelian randomization suggested there was no association with CHD risk. Furthermore, the Mendelian randomization estimate of the component of BMI predicted by the variants in Cluster 5 was negative. That is, in Cluster 5, we have identified genetic variants related to a BMI increasing pathway that is protective of CHD. Orientating to the BMI-increasing alleles, these genetic variants are associated with a favourable metabolic profile, namely increased HDL and decreased SBP, triglycerides, WHR and type 2 diabetes liability.

By analysing the biological pathways underpinning the different clusters, we found evidence supporting that the heterogeneity between the effects of the different components of BMI on cardiovascular risk may be related to inflammation. Furthermore, our findings identify possible inflammatory pathways related to elevated BMI that represent therapeutic targets for preventing CHD. Specifically, the estimated effects of Cluster 5 variants, in contrast to the BMI increasing variants more generally, are consistent with lower levels of key inflammatory cytokines implicated in CHD pathogenesis, including HGF [35], MCP1 [36] and TRAIL [37]. By ameliorating the increased inflammation attributable to elevated BMI, its detrimental effects on CHD risk may also be mitigated.

A number of studies have previously sought to identify genetic variants associated with metabolically favourable adiposity. Huang et al. [38] conducted pairwise significance tests between adiposity traits and various other cardiometabolic traits to identify genetic variants which, for at least one such pairing, associate with an increase in the adiposity trait and a decrease in the cardiometabolic trait. A similar approach to identifying genetic variants associated with favourable adiposity has also been performed by Yaghootkar et al. [39]. Our approach differs to these in that our clusters are formed without using genetic associations with the risk factor or outcome of interest, in this case BMI and CHD, but rather in relation to the chosen traits. Therefore, any difference between clusters in their associations with CHD risk is a meaningful statistical test, rather than a difference driven by the clustering algorithm.

The proposed approach has some limitations. It uses as input the full covariance matrix of the genetic variant-trait associations. If it assumed that the traits are uncorrelated or that the genetic variant-trait associations are estimated in separate samples, then these matrices can be easily constructed from the standard errors of the genetic association estimates which are typically available from published GWAS results. In practice, it is unlikely that the entire set of traits will be uncorrelated, since they would typically be related at least via common association with the primary trait of interest. We have shown how the full covariance matrices can be estimated using estimates of the trait correlations, either from individual level data or from a reference dataset. Furthermore, the simulation study suggested that, unless the traits are highly correlated with each other, the method is robust to ignoring the genetic variant-trait association correlations. This also suggests that the approach is robust to some participant overlap in the samples. If the traits are highly correlated, there is significant sample overlap, and individual level data are not available, there exist methods to estimate the correlation between genetic associations using summary level data. One approach is to use the intercept term from cross-trait LD score regression [40]. Another is to estimate the correlation between genetic association estimates using only variants which are assumed to not be associated with the traits [41].

Another limitation is that the results are dependent on the choice of traits used to cluster on. Domain knowledge should be used to select a set of traits which are believed to be informative of potential mechanisms of the genetic variants under consideration. Future research will look to extend the method to include feature selection [42], so that the inclusion of a moderate to large number of traits, many of which may not distinguish between clusters, is possible. It should be noted that adding highly correlated traits does not add much extra information, and may impact the results if correlation estimates are not incorporated. Thus, if there are a number of traits of interest which are highly correlated, it is better to choose just one of them.

In the applied example, the genetic variants used for clustering were chosen according to them being associated with a primary trait of interest, in this case BMI. This resulted in a fairly large number of variants to cluster, in part because of the very large sample size of the GWAS in which these associations were estimated. Other traits of interest may not have so many independent variants associated with them at genome-wide significance. A low number of variants may make it more difficult to find true clusters if the cluster sizes are small. Nonetheless, there are many traits for which, say, 100 or more variants have been found to associate, and this will only grow as GWAS sample sizes increase. Furthermore, the simulation results showed that our clustering approach is still generally able to detect relatively small clusters, with clusters as small as 10 variants out of 100 in total in some settings. In the case where there are only a very small number of variants associated with the primary trait of interest, we would recommend lowering the threshold for inclusion below genome-wide significance rather than include correlated variants. Genetic variants which are not independent would be expected to associate similarly with the given traits, and so it would not be informative to include these.

In conclusion, we have presented a procedure for clustering genetic variants based on their direction of association with relevant traits, in order to gain insight into their underlying biological mechanisms and pathways. We have demonstrated the utility of clustering genetic variants in this way by applying the method to BMI associated genetic variants and performing Mendelian randomization analyses to infer the differential effects of distinct BMI increasing pathways on CHD risk.

## Methods

### The von Mises–Fisher distribution

The $m$-dimensional von Mises–Fisher (vMF) distribution has probability density function

$$f(\boldsymbol{x} \mid \boldsymbol{\mu}, \kappa) = C_m(\kappa)e^{\kappa \boldsymbol{\mu}' \boldsymbol{x}},$$

where $\|\boldsymbol{x}\| = \|\boldsymbol{\mu}\| = 1$ and $C_m(\kappa)$ is a normalising constant given by

$$C_v(x) = \frac{x^{v/2-1}}{(2\pi)^{v/2}I_{v/2-1}(x)},$$

where $I_v(x)$ is the modified Bessel function of the first kind and order $v$ [12, 43]. The mean parameter $\boldsymbol{\mu}$ is a unit vector which represents the direction from the origin in $m$-dimensional space. The concentration parameter $\kappa$ represents the spread of observations around the mean. When $\kappa = 0$, the distribution is the uniform distribution on the $(m-1)$-dimensional unit sphere. As $\kappa$ increases, the distribution becomes increasingly focused around the point on the unit sphere given by $\boldsymbol{\mu}$.

## The noise-augmented von Mises–Fisher mixture model

Suppose we have $m$-dimensional observations $\{\boldsymbol{x}_1, \ldots, \boldsymbol{x}_n\}$ where $\|\boldsymbol{x}_j\| = 1$ for all $j$ (if the observations are not normalised to have magnitude 1, then this normalisation is the first step in the procedure). Here, $x_j$ represents the vector of proportional association estimates for genetic variant $j$ with the $m$ traits. That is, if standardised genetic association estimates are being used, the vector $\hat{\boldsymbol{\Sigma}}_j^{-1/2}\hat{\boldsymbol{\beta}}_{j\cdot}$ is normalised to have magnitude 1. Further suppose that each observation either belongs to one of $K$ clusters, each cluster containing observations from a vMF distribution, or else belongs to none of these clusters and is therefore considered noise. We can represent this with the $K + 1$ component vMF mixture model given by

$$p(\boldsymbol{x}_j \mid \boldsymbol{\Theta}) = \sum_{k=1}^{K+1} p(\boldsymbol{x}_j, z_j = k \mid \boldsymbol{\mu}_k, \kappa_k) = \sum_{k=1}^{K+1} \pi_k f(\boldsymbol{x}_j \mid \boldsymbol{\mu}_k, \kappa_k)$$

for the $j$th observation, where:

- $\boldsymbol{\Theta} = \{\boldsymbol{\mu}_1, \ldots, \boldsymbol{\mu}_K, \kappa_1, \ldots, \kappa_K, \pi_1, \ldots, \pi_{K+1}\}$;
- $\boldsymbol{z} = \{z_1, \ldots, z_n\}$ denotes cluster membership (that is, $z_j = k$ if $\boldsymbol{x}_j$ belongs to cluster $k$);
- $\pi_k$ is the mixing proportion of cluster $k$, with $\sum_{k=1}^{K+1} \pi_k = 1$;
- $f(\boldsymbol{x}|\boldsymbol{\mu}, \kappa)$ is the density function of the $m$-dimensional vMF distribution;
- $\boldsymbol{\mu}_{K+1}$ is the unit vector which is fixed according to the global sample mean direction, given by

$$\boldsymbol{\mu}_{K+1} = \frac{\sum_{j=1}^n \boldsymbol{x}_j}{\| \sum_{j=1}^n \boldsymbol{x}_j \|};$$

- $\kappa_{K+1}$ is fixed at a number close to zero (for example 0.0001).

In this model, cluster $K + 1$ is referred to as the noise cluster. With $\kappa$ close to zero, the distribution function represents the uniform distribution on the $(m - 1)$-dimensional unit sphere, and so observations which do not fit well to the other $K$ clusters will tend to be assigned here. Note that, since the noise cluster is uniformly distributed, the value of $\boldsymbol{\mu}_{K+1}$ is arbitrary, and we choose the global sample mean for convenience. The use of a uniform distribution for a noise cluster has been commonly used in Gaussian mixture models [44], and our model gives a directional analogue of this approach. Alternative approaches to incorporating a noise component to Gaussian mixture models have also been proposed [45–47]. Although beyond the scope of the present work, different noise distributions for NAvMix could be explored by changing the density of component $K + 1$.

The log-likelihood function is

$$l_K(\boldsymbol{\Theta}) = \sum_{j=1}^n \log\left\{ \sum_{k=1}^{K+1} \pi_k f(\boldsymbol{x}_j \mid \boldsymbol{\mu}_k, \kappa_k) \right\}.$$

In order to maximise the likelihood function to obtain estimates of the parameters $\boldsymbol{\Theta}$, we would require knowledge of the latent variables $\boldsymbol{z}$. Mixture models of this sort are thus fitted using the EM algorithm [48].

**The EM algorithm.** Suppose we have an estimate of $\boldsymbol{\Theta}$, denoted by $\hat{\boldsymbol{\Theta}}$. Let $Q(\boldsymbol{\Theta} \mid \hat{\boldsymbol{\Theta}}) = E_{z|X,\hat{\boldsymbol{\Theta}}} l_K(\boldsymbol{\Theta})$. Then

$$Q(\boldsymbol{\Theta} \mid \hat{\boldsymbol{\Theta}}) = \sum_{j=1}^{n} \sum_{k=1}^{K+1} \gamma_{jk} \log\{\pi_k f(\boldsymbol{x}_j \mid \boldsymbol{\mu}_k, \kappa_k)\},$$

where

$$\gamma_{jk} = \Pr\left(z_j = k \mid \boldsymbol{x}_j, \hat{\boldsymbol{\Theta}}\right) = \frac{\pi_k f(\boldsymbol{x}_j \mid \boldsymbol{\mu}_k, \kappa_k)}{\sum_{l=1}^{K+1} f(\boldsymbol{x}_j \mid \boldsymbol{\mu}_l, \kappa_l)}, \quad k = 1, \ldots, K+1.$$

Computing the $\gamma_{jk}$ for a given $\hat{\boldsymbol{\Theta}}$ is the E step in the EM algorithm.

Given the $\gamma_{jk}$, we can estimate $\boldsymbol{\Theta}$ by maximising $Q(\boldsymbol{\Theta} \mid \hat{\boldsymbol{\Theta}})$. Following Banerjee et al. [12], the parameter estimates are obtained from

$$\hat{\boldsymbol{\mu}}_k = \frac{\sum_{j=1}^{n} \gamma_{jk} \boldsymbol{x}_j}{\| \sum_{j=1}^{n} \gamma_{jk} \boldsymbol{x}_j \|}, \quad k = 1, \ldots, K,$$

$$\frac{I_{m/2}(\hat{\kappa}_k)}{I_{m/2-1}(\hat{\kappa}_k)} = \frac{\| \sum_{j=1}^{n} \gamma_{jk} \boldsymbol{x}_j \|}{\| \sum_{j=1}^{n} \gamma_{jk} \|}, \quad k = 1, \ldots, K \qquad (1)$$

$$\hat{\pi}_k = \frac{1}{n} \sum_{j=1}^{n} \gamma_{jk}, \quad k = 1, \ldots, K+1.$$

This is the M step of the EM algorithm. Note that we do not update the noise cluster parameters, $\boldsymbol{\mu}_{K+1}$ and $\kappa_{K+1}$, but we do update the proportion of observations which are assigned to the noise cluster, $\hat{\pi}_{K+1}$. Now, (1) does not give a closed form solution for computing $\hat{\kappa}_k$. However, a number of methods for approximating these solutions have been proposed which allow the concentration parameter estimates to be easily updated. Banerjee et al. [12] proposed the approximation

$$\hat{\kappa}_k = \frac{\bar{r}_k m - \bar{r}_k^3}{1 - \bar{r}_k^2},$$

where

$$\bar{r}_k = \frac{\| \sum_{j=1}^{n} \gamma_{jk} \boldsymbol{x}_j \|}{\| \sum_{j=1}^{n} \gamma_{jk} \|}.$$

Hornik and Grün [15] summarise several other approximation methods and provide software for implementing each of them. Note that, in practice, values of $\bar{r}$ very close to 1 can cause numerical problems (due to the fact that this relates to the case where the observations are almost all at the same point, and the precision is thus close to infinity). To get around this, we cap the value that $\hat{\kappa}_k$ can take at 500.

The EM algorithm can be started at either the E step, given an initial estimate of $\boldsymbol{\Theta}$, or at the M step, given initial values of the $\gamma_{jk}$. The algorithm is iterated until the absolute value of the difference between successive values of $l_K(\hat{\boldsymbol{\Theta}})$ is less than some predefined convergence threshold. In our simulation study and applied example, we used $10^{-4}$ as the convergence threshold.

**Initialisation of the algorithm.** In order to initialise the algorithm, we must first set an initial proportion of observations which belong in the noise cluster, which we will denote by

$0 < \hat{\pi}_{K+1}^{(0)} < 1$. We then perform the spherical k-means procedure [14], which clusters observations based on similarity of their direction from the origin, analogous to the k-means procedure which clusters observations based on Euclidean distance. We take as initial values, for $i = 1, \ldots, n$,

$$
\gamma_{ik} = \begin{cases} 1 - \hat{\pi}_{K+1}^{(0)}, & \text{if observation } i \text{ is assigned to cluster } k \\ 0, & \text{otherwise} \end{cases}, \quad k = 1, \ldots, K
$$

$$
\gamma_{i(K+1)} = \hat{\pi}_{K+1}^{(0)}.
$$

We then begin the EM algorithm at the M step. Note that the spherical k-means procedure relies on an initial random set of cluster means, and thus its results are sensitive to this randomisation. There is a possibility that certain initial values from the procedure will result in the EM algorithm converging to a local, rather than global, maximum. We therefore run the algorithm a number of times in practice, each time beginning with different initial values. We take as final parameter estimates those which result in the EM algorithm converging to the greatest maximum. In our simulation study and applied example, we ran the algorithm with 5 different initialisations.

**Choosing the number of clusters.** In practice, we will not know the number of clusters to fit to the data. The number of clusters can be determined using an information criterion, for example BIC [44, 49]. For successive values of $K$, we perform the algorithm above and compute

$$
\phi_m(K) = -2l_K(\hat{\Theta}) + r_m(K) \log(n),
$$

where $r_m(K) = (m + 2)K + m$ is the number of parameters estimated. We continue until $\phi_m(K)$ increases for successive iterations. The final number of clusters is then taken to be $\arg\min_K \phi_m(K)$.

**Assigning cluster membership.** Output from the procedure for fitting the mixture model is a set of probabilities for each observation belonging to each cluster (that is, the $\gamma_{ik}$ parameters). The simplest approach for assigning cluster membership is to assign each observation to the cluster for which it has the greatest probability of membership (that is, $\hat{z}_i = \arg\max_k \gamma_{ik}$). This is the approach used in both the simulation study and the applied example presented in this paper.

Mixture model approaches to clustering allow for flexibility in the way that cluster membership is assigned. For increased confidence in the clusters, a threshold could be set such that an observation is only assigned to a cluster if the probability of membership is greater than a certain level. Those which do not meet the threshold for any cluster remain unassigned. Finally, soft clustering is possible, whereby observations are assigned to any cluster for which its probability of membership is greater than a certain level. Under the soft clustering approach, an observation may be assigned to more than one cluster.

### Genetic variant-trait association covariance matrix

For variant $j$, the $(k, l)$th element of $\hat{\Sigma}_j$ is given by

$$
\text{se}(\hat{\beta}_{jk})\text{se}(\hat{\beta}_{jl})\text{cor}(\hat{\beta}_{jk}, \hat{\beta}_{jl}),
$$

where $\text{se}(\hat{\beta}_{jk})$ is the standard error of $\hat{\beta}_{jk}$. If the genetic variant-trait associations are estimated in separate, non-overlapping, samples, then $\text{cor}(\hat{\beta}_{jk}, \hat{\beta}_{jl}) = 0$ and $\hat{\Sigma}_j$ can be taken to be the

diagonal matrix with $k$th diagonal entry equal to $\mathrm{se}^2(\hat{\beta}_{jk})$. If the traits are estimated in the same sample, then the off-diagonal entries of $\hat{\Sigma}_j$ will be non-zero. Although the correlation between $\hat{\beta}_{jk}$ and $\hat{\beta}_{jl}$ is not easily estimated, provided the $j$th genetic variant explains only a small proportion of the variance in the $k$th and $l$th traits, then $\mathrm{cor}(\hat{\beta}_{jk}, \hat{\beta}_{jl}) \approx \mathrm{cor}(X_k, X_l)$, where $X_k$ and $X_l$ are the $k$th and $l$th traits, respectively [50]. We can therefore compute the $(k, l)$th entry of $\hat{\Sigma}_j$, $i \neq j$, by

$$\mathrm{se}(\hat{\beta}_{jk})\mathrm{se}(\hat{\beta}_{jl})\widehat{\mathrm{cor}}(X_k, X_l),$$

where $\widehat{\mathrm{cor}}(X_k, X_l)$ is an estimate of the correlation between $X_k$ and $X_l$. As a result of this, if the traits are assumed to be independent, then the off-diagonal entries of $\hat{\Sigma}_j$ can be approximated by zeros, and the covariance matrix taken to be diagonal as in the separate samples case.

## Simulation study

We simulated $n = 100$ independent genetic variants for $N = 20000$ individuals, denoted $G_{ij}$ for individual $i$ and genetic variant $j$, and $m$ traits, denoted $X_{il}$ for individual $i$ and trait $l$, from the following model

$$\mathrm{maf}_j \sim \mathrm{Uniform}(0.01, 0.5)$$
$$G_{ij} \sim \mathrm{Binomial}(2, \mathrm{maf}_j)$$
$$U_i, \varepsilon_{i1}, \dots, \varepsilon_{im} \sim N(0, 1), \text{ independently}$$
$$L_{ik} = \sum_{j \in n^{(k)}} \beta_{jk} G_{ij}$$
$$X_{il} = \sum_{k=1}^{K} \delta_{kl} L_{ik} + \sum_{j \in n^{(K+1)}} \alpha_j G_{ij} + \gamma U_i + \sqrt{1 - \gamma^2}\varepsilon_{il},$$

for $i = 1, \dots, N$ and $l = 1, \dots, m$. The variables $L_1, \dots, L_K$ are latent factors which represent $K$ different mechanisms by which the genetic variants act on the observed traits $X_1, \dots, X_m$, with $n^{(k)}$ indexing the variants which are associated with $L_k$. The variants indexed by $n^{(K+1)}$ are those in the noise cluster. These variants act directly on the traits and do not associate with any of the latent factors. The common variable $U_i$ induces correlation between the traits, with the amount of correlation determined by $\gamma$. The relationship between the genetic variants in the $k$th cluster and the other variables are illustrated in the directed acyclic graph in Fig G in S1 Text. The number of traits was either $m = 2$ or $9$ and we set $\gamma = 0, 0.4$ or $0.8$. The first 80 variants were split into 1, 2 or 4 clusters, with the remaining 20 variants considered to be noise. For the $k = 2$ scenarios, each cluster contained 40 variants. For the $k = 4$ scenarios, the cluster sizes were 30, 20, 20 and 10.

We generated the $\beta_{jk}$ values such that most of the genetic variants were weakly associated with the traits, while a relatively small number of them were associated more strongly. For each $k$, and for each $j \in n^{(k)}$, with probability $1 - \phi$, $\phi \sim \mathrm{Uniform}(0.05, 0.2)$, $\beta_{jk}$ was generated from the Uniform$(0.03, 0.06)$ distribution (which results in a p-value, on average, below the genome-wide significance level), and with probability $\phi$ from the $N(0.1, 0.02^2)$ distribution. For $j \notin n^{(k)}$, $\beta_{jk}$ was set to zero. The $\alpha_j$ values were generated from the Uniform$(-0.1, 0.1)$ distribution, $j \in n^{(K+1)}$, and set to zero otherwise.

When $m = 2$, $\delta_{kl}$ was set to the $(k, l)$th element of the matrices

$$
(1 \quad 1), \quad \begin{pmatrix} 1 & 1 \\ 1 & -1 \end{pmatrix}, \quad \begin{pmatrix} 1 & 1 \\ 1 & -1 \\ -1 & 1 \\ -1 & -1 \end{pmatrix},
$$

for the 1, 2 and 4 cluster scenarios, respectively. When $m = 9$, $\delta_{kl}$ was set to the $(k, l)$th element of the matrices

$$
(1 \quad 1 \quad 1 \quad 1 \quad 1 \quad 1 \quad 1 \quad 1 \quad 1),
$$

$$
\begin{pmatrix} 1 & 1 & 1 & 0.5 & 0.5 & 0.5 & 0.5 & 0.5 & 0.5 \\ 1 & -1 & -1 & 0.5 & 0.5 & 0.5 & 0.5 & -0.5 & -0.5 \end{pmatrix},
$$

$$
\begin{pmatrix} 1 & 1 & 1 & 0.5 & 0.5 & 0.5 & 0.5 & 0.5 & 0.5 \\ 1 & 1 & 1 & 1 & 1 & 0 & 0 & 0 & 0 \\ -1 & -1 & -1 & 0.5 & 0.5 & 0.5 & 0.5 & -0.5 & -0.5 \\ -1 & -1 & -1 & -1 & -1 & 0 & 0 & 0 & 0 \end{pmatrix},
$$

for the 1, 2 and 4 cluster scenarios, respectively. These values determine the direction and relative magnitude of association between the genetic variants in each cluster with the traits. For example, in the $m = 2$, $K = 2$ scenario, one cluster contains variants which are positively associated with both traits, whereas the other cluster contains variants that are positively associated with trait 1 and negatively associated with trait 2. The parametrisation of the $\alpha_j$, $\beta_{jk}$ and $\delta_{kl}$ parameters are such that the proportion of variance of each trait explained by the genetic variants was approximately 5–10%.

The estimated genetic variant-trait associations were computed using simple linear regression of each trait on each genetic variant in turn. The resulting datasets were clustered using NAvMix with an initial proportion of genetic variants in the noise cluster of 0.05, and using mclust with an initial noise cluster of of 5 randomly selected genetic variants.

A supplementary simulation study was also performed where the sample size differed for each trait. Each sample size was randomly chosen to be between 10000 and 50000. The results of this supplementary simulation study is presented in S1 Text.

## Clustering BMI associated genetic variants

Genetic variant association estimates with BMI were taken from the GWAS of Pulit et al. [19]. Variants with p-value $< 5 \times 10^{-8}$ were pruned using the TwoSampleMR package in R [51] with $r^2 = 0.001$.

Genetic variant association estimates with body fat percentage, SBP, triglycerides and HDL were taken from results from the Neale Lab which are based on the UK Biobank dataset (http://www.nealelab.is/uk-biobank/). Genetic variant associations for educational attainment were taken from the GWAS of Okbay et al. [52]; for physical activity, the GWAS of Doherty et al. [53]; for lifetime smoking score, the GWAS of Wootton et al. [54]; for WHR the GWAS of Pulit et al. [19]; and for type 2 diabetes, the GWAS of Mahajan et al. [6]. Note that for the educational attainment dataset, one BMI associated genetic variant (rs10761785) was replaced

with a proxy (rs2163188) with $r^2$ = 0.9842 (identified using PhenoScanner [55, 56]). All studies used were performed on samples of individuals of European ancestry or predominantly European ancestry. All genetic variant trait-association estimates were orientated with respect to the alleles such that the associations with BMI were positive. Table E in S1 Text shows the sample sizes for each study as well as the number of the BMI associated genetic variants which associate with each trait at the genome-wide significance level.

Clustering was performed using NAvMix with an initial proportion of genetic variants in the noise cluster of 0.05, and 5 separate initialisations of the algorithm was used. The probability of membership of each genetic variant to each cluster produced by the algorithm is shown in S1 Table.

## Mendelian randomization analyses

A genetic variant is a valid instrumental variable for a Mendelian randomization analysis if it is: associated with the risk factor; independent of any confounders of the risk factor-outcome relationship; and has no causal pathway to the outcome other than via the risk factor [57]. Under the two-sample framework, the genetic variant-risk factor and genetic variant-outcome associations are estimated in separate samples [24]. Under the assumption that all variants in the analysis are valid instruments, MR-IVW produces a statistically consistent estimator of the causal effect and a test for the causal null hypothesis [25]. The three methods used for sensitivity analyses were chosen since they each produce a valid estimate of the causal effect of BMI on CHD under different assumptions [58]: MR-Median (a majority of the genetic variants are valid instrument); the Contamination Mixture method (a plurality of the genetic variants are valid instruments); and the MR-PRESSO method (the InSIDE assumption is met). The intercept test from the MR-Egger method was used to test for the presence of unmeasured directional pleiotropy. Analyses were carried out using the MendelianRandomization [59, 60] and MRPRESSO [28] packages.

Genetic variant association estimates with CHD were taken from the CARDIoGRAMplusC4D dataset of Nikpay et al. [61] and accessed using PhenoScanner [55, 56]. Genetic variant associations with CRP were taken from results from the Neale Lab which are based on the UK Biobank dataset (http://www.nealelab.is/uk-biobank/). Genetic variant association estimates with the 41 cytokines and growth factors were taken from the data supporting Ahola-Olli et al. [31] and Kalaoja et al. [32]. Table F in S1 Text gives a list of the BMI associated genetic variants which were not available in each of the outcome datasets and were therefore excluded from the relevant Mendelian randomization analyses.

## Gene mapping and gene set analysis

The 539 BMI associated genetic variants were mapped to genes using the SNP2GENE function in FUMA [30]. Summary statistics for each cluster of variants were uploaded separately, and were identified as pre-defined lead SNPs. Both positional and eQTL mapping was performed. For the eQTL mapping, tissue types were selected as all those from the following sources: EQTL catalogue; PsychENCODE; van der Wijst et al. scRNA eQTLs; DICE; eQTLGen; Blood eQTLs; MuTHER; xQTLServer; ComminMind Consortium; BRAINEAC; and GTEx v8. All other default settings were used. Gene set analysis was performed using the GENE2FUNC function. The results presented in S2 Table include all canonical pathways from MsigDB, as well as gene ontology processes, which associate with the mapped genes using hypergeometric tests (with multiple test correction applied per cluster).

## Supporting information

**S1 Text. Additional simulation results and supplementary information for the simulation study and applied example.**
(PDF)

**S1 Table. Allocated cluster and probability of membership to each cluster for each BMI associated genetic variant.**
(XLSX)

**S2 Table. List of canonical pathways and gene ontology processes associated with the mapped genes for each cluster of BMI associated genetic variants.**
(XLSX)

**S3 Table. Results from Mendelian randomization sensitivity analyses of the effect of BMI on cytokines and growth factors.** Estimates and 95% confidence intervals from MR-Median, the Contamination Mixture method (MR-ConMix) and MR-PRESSO for the association of genetically predicted BMI with 41 cytokines and growth factors.
(XLSX)

## Author Contributions

**Conceptualization:** Stephen Burgess.

**Formal analysis:** Andrew J. Grant.

**Methodology:** Andrew J. Grant, Dipender Gill, Paul D. W. Kirk, Stephen Burgess.

**Software:** Andrew J. Grant.

**Visualization:** Andrew J. Grant.

**Writing – original draft:** Andrew J. Grant, Stephen Burgess.

**Writing – review & editing:** Andrew J. Grant, Dipender Gill, Paul D. W. Kirk, Stephen Burgess.

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
