## [Decision Letter · Decision Letter 0]

17 Jul 2021

Dear Dr Grant,

Thank you very much for submitting your Research Article entitled 'Noise-augmented directional clustering of genetic association data identifies distinct mechanisms underlying obesity' to PLOS Genetics.

The manuscript was fully evaluated at the editorial level and by independent peer reviewers. The reviewers appreciated the attention to an important problem, but raised some substantial concerns about the current manuscript. Based on the reviews, we will not be able to accept this version of the manuscript, but we would be willing to review a much-revised version. We cannot, of course, promise publication at that time.

If you decide to revise the manuscript for further consideration at PLOS Genetics, please aim to resubmit within the next 60 days, unless it will take extra time to address the concerns of the reviewers, in which case we would appreciate an expected resubmission date by email to plosgenetics@plos.org.

[LINK]

We are sorry that we cannot be more positive about your manuscript at this stage. Please do not hesitate to contact us if you have any concerns or questions.

Yours sincerely,

Michael P. Epstein

Associate Editor

PLOS Genetics

David Balding

Section Editor: Methods

PLOS Genetics

Reviewer's Responses to Questions

**Comments to the Authors:**

Reviewer #1: The manuscript by Grant and colleagues proposed a novel method for clustering genetics variants based on their associations with multiple related traits. The advantage of cross-trait analysis, compared with single-trait analysis, is that it can potentially partition variants underlying a trait of interest into multiple clusters/patterns, each associated with a distinct mechanism. The manuscript proposed a novel approach to variant clustering, based on directions of variant effects on multiple traits. This is an interesting and well-motivated idea and is a timely contribution to this field. Applying this method, NAvMix, the authors reported some interesting findings in BMI and related traits.

Major comments

(1) One major issue of cross-trait analysis is overlapping samples between different studies. This is potentially a serious problem, as it would induce correlations of variant effects across traits, a pattern potentially mis-identified as true clusters. The proposed strategy to deal with this seems inadequate. Most often, individual level data is not available, and according to the authors, NAvMix, will simply set the off-diagonal terms in the covariance matrix 0, i.e. ignoring the possible correlations. The justification is that this is not a problem in simulations. However, it's possible that in the simulation setting, the correlations induced by shared samples are too weak, compared with the correlation of true effect sizes. As a result, ignoring induced correlations does not change the results much. This may not be the case, however, in the real data.

My suggestion is: better justification using more extensive simulations. In particular, it would be helpful to vary the relative importance of true effect size correlations vs. correlation induced by shared samples. It would be helpful, for example, to vary how much phenotypic variations are explained by genetic variants vs. by the shared underlying factors ($U_i$, in simulations). Related to this point: it would be helpful to better explain some of the parameter settings in simulations. Ex. effect sizes are drawn from certain distributions. Are the parameters of these distributions realistic?

Another suggestion: it is possible to estimate correlation of effect sizes between traits, due to shared samples, using cross-trait LD score regression [PMID: 26414676]. That paper uses full GWAS summary statistics to estimate genetic correlation, while accounting for shared samples. It shows that the "residual" correlation due to shared samples is a simple function of phenotypic correlations, the sample sizes of the two studies, and the number of shared samples. In fact, it can directly estimate this residual correlation. It may be possible to directly use these estimates (at an appropriate scale) in NAvMix.

(2) The noise cluster presumably increases the robustness of the method. However, in the real data, only 1 out of >500 BMI variants is assigned to the noise cluster. One possible explanation is that: the mean of the noise cluster is the global mean direction. This may be too similar to the major clusters (clusters 1 and 2 in the BMI results). As a result, not many variants would be assigned to the noise cluster. It seems to make sense to use a broader distribution (uniform at some scale) for the noise cluster.

(3) The simulation procedure largely follows the model of NAvMix. So perhaps unsurprisingly, NAvMix performs well under those simulations. A more realistic, or biologically motivated simulation would give a better idea of the performance of NAvMix. The authors could explicitly simulate a few "latent factors", which act on the observed phenotypes. Then each variant could either act on these latent fators, or act on some traits directly. I think if each trait acts only on one factor a time, then this reduces to the variant clustering problem. The advantage of this simulation regime is that one could easily vary some of the parameters and assess if NAvMix are robust to these changes. For example, is the method robust to direct effects of variants (i.e. variants acting on a trait without affecting any of the factors)?

(4) The main difference of clusters 1 and cluster 2 is that cluster 1 variants are associated with smoking, but cluster 2 not. I found this result interesting/puzzling. Given the smoking association, I would imagine that cluster 1 may capture some "behavior" component of BMI. Indeed, it was reported that hertiability of obesity is mostly enriched in brain, and it is not hard to imagine that smoking variants act through some kind of additive behavior. However, the pathway analysis of cluster 1 does not suggest anything related to behavior/brain. One concern is that the results may be driven by sample sharing between BMI and smoking GWAS, though it's not clear to me if this would lead to different clusters. In any case, it would be helpful to check this - could be done using the intercept term of cross-trait LDSC, as mentioned above. In fact, it would be good to check the correlations driven by shared samples for all pairs of traits.

Another relevant question is: has BMI GWAS data adjusted smoking, and vice versa? Is it possible that associations of BMI variants with smoking driven by some kind of collider bias? This could happen when one has a causal model like this: SNP -> smoking <- additive behavior -> BMI. Adjusting smoking in GWAS of BMI may lead to false association of SNP and BMI.

(5) The authors suggest that cluster 5 variants represent some kind of "favorable adiposity", that reduces inflammatory cytokines and hence the CHD risk. This is an attractive model. However, I would caution against over-interpretation. The data basically says that cluster 5 variants have effects on inflammatory cytokines. It's unclear that this effect is mediated through favorable/protective adiposity. It's possible that these variants have some pleiotropic effects on both BMI and cytokines, without adoposity playing any role in the regulation of these cytokines. To rigorously make this claim, I'd imagine some mediation type of analysis, where one controls BMI or some measures of favorable adiposity, and show that the cluster 5 variants would be no longer associated with the CHD risk. I doubt such data is available for this type of analysis though.

Minor comments

- Variants in cluster 4 have metabolic functions, and are associated with HDL and TG. However, it is not found to have an effect on CHD risk in MR analysis. This is a bit unexpected, given that LDL is a known risk factor of CHD. It would be good to examine the LDL association of cluster 4 variants.

- Typo: line 314, norm of $x_j$ = 1 for all $i$ - I suppose it should be "for all j".

- Typo: denoted $X_{ik}$ for individual i and trait l, in line 406.

- Table S5: it would be helpful to show the fold enrichment (or expected number of overlapping genes). Also, the authors used KEGG and REACTOME. It would be helpful to add GO analysis as well, which is more commonly used for this type of analysis, and may cover more biological processes.

Reviewer #2: The authors have proposed a new procedure NAvMix (and its R implementation) to cluster variants based on their genetic effects on multiple traits. The procedure considers a separate cluster for noise on top of the existing clustering method on the unit hypersphere based on a mixture of von Mises-Fisher distributions. The authors have shown good performance of NAvMix in simulations. The authors have applied NAvMix on genetic variants associated with BMI to unravel distinct biological pathways. However, I have some concerns with the results. The simulation study is somewhat narrow in scope (e.g. their data generating mechanism appears close to the main feature of their procedure that they are leveraging; a wider range of evaluation metrics could have been considered). Takeaways from different real data analyses are not always clear. I have provided detailed comments below.

**Major Comments**

1. “Assuming all genetic associations are estimated with the same sample size for a given trait, this will not distort the direction vector. Otherwise, the direction vector will be weighted toward traits for which the associations are more precisely estimated.” - Is this a limitation of the method? What would be practical solution to this issue? Are the authors recommending using NAvMix on traits with similar sample size only? When working with summary statistics on different traits, it is often the case that sample sizes differ widely. For e.g. recent meta-analysis [PMID 34059833] of glycemic traits included ~90K samples for 2hr glucose whereas >280K samples for fasting glucose.

2. Lines 68-69: Genetic variants for which vector of association estimates are input in the algorithm are assumed to be independent of each other. Do the authors recommend that only the top hit in a locus be included in the set to be clustered? But then, for most traits, only a handful of loci are significant (not anywhere close to the number 100 assumed in the simulations or the 539 variants in the their real data analysis— a huge number that one can get only from very large sample sizes like they have used). Can this method be used for maybe 10-20 independent variants or on traits with modest sample sizes? Do the authors recommend a lower limit to these numbers?

3. The data generating mechanism (DGM) for the simulations seems somewhat favorable for the “directional statistics” concept that the authors are leveraging in their procedure. In my opinion, it will be useful to consider more realistic and a different DGM. For e.g., instead of using angles between genetic effects to generate different clusters of effects, the authors can consider a structural equation model (as their DGM) reflecting different DAGs capturing the different underlying biological mechanisms among traits (for different DAGs the authors may follow Fig 1 of PMID 29226067)

4. Clustering evaluation metrics: The authors considered Rand Index. Isn’t adjusted Rand index a better metric to use since it is adjusted for chance? What about additionally using other popular metrics (like normalized mutual information, Silhouette index, etc) that capture other aspects of clustering that Rand index cannot? For evaluation, I think considering metrics that do not require a priori knowledge of clusters is also important.

5. Fig 5: I see that different clusters of genetic instruments may or may not give different causal relation result. I don’t see a clear distinction between clusters in terms of their MR results. Can the authors clarify the takeaway from the MR analysis done here?

6. The required input/variables/format for their R package is not clear. What parameter choices do users need to make when using NAvMix package? It will be useful for readers if the authors can provide an outline of how readers can implement their procedure from start to finish.

**Minor Comments**

7. Line 149: Are the authors including SNPs with r^2<0.001? If not, I am not sure why Lines 68-69 say that variants should not be in LD.

8. Lines 332-333: specify the contents of parameter vector \\Theta.

9. There might be some notational inconsistency here and there. For e.g., what do X_j, X_k stand for in Line 398 and are they related to vector x_j in the previous paragraphs? Is vector x_j =\\hat\\Sigma_j^{-1/2} \\hat\\beta_j? It is worth clarifying these relations for readers.

10. What happens to \\pi_k estimate if there are only a few members in a cluster (other than the noise cluster)? Will the estimate be unstable? Can unstable estimates for such a cluster lead to lack of convergence?

11. Figs 3-7: It will be helpful to have the number of SNPs in each cluster mentioned in the figures.

12. Figs 6 & 7: Only MR-IVW results are presented. MR-IVW is least likely to be robust to violation of assumptions (it is unlikely that all 539 SNPs are valid instruments). Is only MR-IVW presented because the results for these MR analyses using different methods/assumptions give the same qualitative results?

Reviewer #3: Noise-augmented directional clustering of genetic association data identifies distinct mechanisms underlying obesity

This paper aims to answer the question ‘how do genetic variants influence complex traits?’ and presents a new approach for clustering genetic variants according to the direction of association with complex traits and incorporates a noise model to improve overall clustering results. They perform simulation studies and apply their method to studying variants associated with high BMI and look at the effects on coronary heart disease.

The paper introduces a novel model but needs more extensive benchmarking and robustness tests to convince me of its utility.

Major Concerns:

I’d like to see some justification/evidence that the magnitude of associations to traits is not important, but the direction of association is the key concern. Is there biological rationale for claiming that magnitudes of association do not encode useful information about how variants influence traits?

Benchmarking should be done over a wider range of comparison models and over parameter values (Why were the number of traits chosen to just be 2 or 9?) I’d also like to see answers to questions about limitations of the model - (How many data points are required to detect clusters reliably? What is the smallest fraction that the method can feasibly detect? How does performance change as a function of noise?)

Moderate Concerns:

I’d like to see a justification for choosing a uniform noise model. Does incorporating this noise model improve performance over simple pre-processing/data cleaning approaches?

I’d like a brief explanation for why they believe that the other approaches they benchmark achieve lower Rand index on their simulations. Why were these models selected for benchmarking?

Is the assumption of no linkage disequilibrium a fair assumption?

Minor Concerns:

Figure 4 is a bit confusing - can this be better illustrated? The figure makes it appear that several clusters are actually quite similar - are the differences between them significant?

Figure 5,6 - What do the dotted lines represent?

**Have all data underlying the figures and results presented in the manuscript been provided?**

Reviewer #1: Yes

Reviewer #2: Yes

Reviewer #3: Yes

PLOS authors have the option to publish the peer review history of their article (what does this mean?). If published, this will include your full peer review and any attached files.

Reviewer #1: No

Reviewer #2: No

Reviewer #3: No

---

## [Decision Letter · Decision Letter 1]

23 Nov 2021

Dear Dr Grant,

Thank you very much for submitting your Research Article entitled 'Noise-augmented directional clustering of genetic association data identifies distinct mechanisms underlying obesity' to PLOS Genetics. We are willing to accept your manuscript once you submit a revised version that addresses the minor comments raised by one reviewer. 

[LINK]

Yours sincerely,

Michael P. Epstein

Associate Editor

PLOS Genetics

David Balding

Section Editor: Methods

PLOS Genetics

Reviewer's Responses to Questions

Reviewer #1: The authors have done a thorough job in addressing all my comments. I do not have any more concerns.

Reviewer #2: The authors have reasonably addressed my previous concerns. Some minor comments below.

1. Table 1: In the caption or somewhere appropriate, it will be useful to provide the actual number of noise variants in the simulated data to better appreciate the estimated “Number of noise variants” from each method.

2. Lines 153-154: The authors say “This suggests that, unless there is substantial trait correlation or sample overlap, the procedure is robust to missing these estimates.” Since the difference in adjusted Rand index between NAvMix and NAvMix (cor) increases quite a bit with increasing number of traits, decreasing number of clusters, and increasing strength of correlation (similar observation for silhouette index in Fig A), the authors need to include these other conditions in the above statement.

3. Regarding this correlation: Authors discuss limitations of NAvMix, which includes “It uses as input the full covariance matrix of the genetic variant-trait associations.” However, I am not sure why this is a limitation. Under reasonable assumptions, the authors approximate \\Sigma_j as in lines 450-451. While individual-level data readily gives Corr(X_k, X_l), one can also estimate this correlation easily and quickly using summary statistics only. The field of cross-phenotype association tests using summary statistics uses such an estimate all the time (see for e.g. PMID 29226385). I believe the authors can use such an estimate instead of making the sub-optimal assumption that off-diagonal elements of \\Sigma_j are 0.

4. I may have missed but did not find details about sample sizes, the number of significant variants in all, and the number of significant variants after pruning for each of the traits considered from Pulit et al, UK Biobank, etc. I think readers will find these details useful in interpreting results from NAvMix. In particular, I think NAvMix requires large sample sizes and highly polygenic traits.

5. With respect to Fig 5, the authors mention “The components of BMI predicted by Clusters 1, 2, and 3 show a similar effect size and direction to the overall result.” Does this mean the clusters 1, 2 and 3 are not quite different and could have been just one cluster?

Reviewer #3: The revision adequately addressed all our concerns

**Have all data underlying the figures and results presented in the manuscript been provided?**

Reviewer #1: Yes

Reviewer #2: None

Reviewer #3: Yes

PLOS authors have the option to publish the peer review history of their article (what does this mean?). If published, this will include your full peer review and any attached files.

Reviewer #1: **Yes: **Xin He

Reviewer #2: No

Reviewer #3: No

---

## [Editor Report · Decision Letter 2]

1 Dec 2021

Dear Dr Grant,

We are pleased to inform you that your manuscript entitled "Noise-augmented directional clustering of genetic association data identifies distinct mechanisms underlying obesity" has been editorially accepted for publication in PLOS Genetics. Congratulations!

Yours sincerely,

Michael P. Epstein

Associate Editor

PLOS Genetics

David Balding

Section Editor: Methods

PLOS Genetics

**Data Deposition**

http://datadryad.org/submit?journalID=pgenetics&manu=PGENETICS-D-21-00715R2

**Press Queries**

---

## [Editor Report · Acceptance letter]

17 Dec 2021

PGENETICS-D-21-00715R2 

Noise-augmented directional clustering of genetic association data identifies distinct mechanisms underlying obesity 

Dear Dr Grant, 

We are pleased to inform you that your manuscript entitled "Noise-augmented directional clustering of genetic association data identifies distinct mechanisms underlying obesity" has been formally accepted for publication in PLOS Genetics! Your manuscript is now with our production department and you will be notified of the publication date in due course.

With kind regards,

Katalin Szabo

PLOS Genetics

On behalf of:
